# Effects of Reablement on the Independence of Community-Dwelling Older Adults with Mild Disability: A Randomized Controlled Trial

**DOI:** 10.3390/ijerph16203954

**Published:** 2019-10-17

**Authors:** Shinji Hattori, Toshiyuki Yoshida, Yasuyuki Okumura, Katsunori Kondo

**Affiliations:** 1Research Department, Institute for Health Economics and Policy, Association for Health Economics Research and Social Insurance and Welfare, Tokyo 105-0003, Japan; hatto@lares.dti.ne.jp; 2Department of Economics, Seijo University, Tokyo 157-8511, Japan; tymk2010@gmail.com; 3Department of Psychiatry and Behavioral Science, Tokyo Metropolitan Institute of Medical Science, Tokyo 156-8506, Japan; 4Center for Preventive Medical Sciences, Chiba University, Chiba 260-0856, Japan; kkondo@chiba-u.jp

**Keywords:** functional limitation, reablement, rehabilitation, long-term care

## Abstract

We aimed to assess the efficacy of a reablement program in improving the independence from long-term care services of older adults with mild disability. This parallel, two-arm, randomized controlled, superiority trial was conducted in Neyagawa, a local government area in Osaka, Japan. Eligible participants were community-dwelling individuals aged ≥65 years certified as support-required level. They were assigned in a 1:1 ratio to receive either a community-based, multicomponent, multidisciplinary, individualized goal-directed, and time-limited intervention (the CoMMIT program) plus standard care or standard care alone. The primary outcome was independence, that is, the nonuse of long-term care services during the three-month follow-up period. The study was terminated early due to slow enrollment. A total of 375 participants were enrolled and randomized to either the intervention (*n* = 190) or control (*n* = 185) group. The proportions of independence were 11.1% and 3.8% in the intervention and control groups, respectively (absolute difference: 7.3; 95% confidence interval: 2.0–12.5). There was no difference in the risk of serious adverse events between the groups. The CoMMIT program plus standard care was found superior to standard care alone in enhancing the independence from long-term care services of older adults with mild disability.

## 1. Introduction

Worldwide, population aging poses challenges for countries in terms of extending “healthy” life expectancy. A national policy for the extension of healthy life expectancy has been implemented in Japan [1], one of the countries with long life expectancy. In Japan, approximately 5% of the individuals aged ≥65 years require long-term care services for recipients with mild disability [2]. Most of them have disabilities in instrumental activities of daily living (IADL), such as shopping for groceries and preparing food [3]. The long-term care services aim to prevent people with disabilities from requiring more caregiving time and help them lead independent daily lives [4]. However, users of these services tend to use them continuously without any improvements in their disabilities.

Improving the independence in daily life of community-dwelling older adults with mild disability is important because disability is associated with further disability progression, institutionalization, hospitalization, and even death [5,6]. Multicomponent interventions are expected to have better effects on enhancing independence than single-component interventions (e.g., a physical activity program alone). However, the efficacy of multicomponent interventions has not been adequately evaluated among community-dwelling older adults with mild disability, to date [7,8,9,10,11,12,13,14,15,16].

Reablement services (termed restorative care in some countries) are approaches to improve the independence of older adults with mild disability [17]. In Japan, reablement services have not yet been accepted as standard care. Hence, we developed a community-based, multicomponent, multidisciplinary, individualized goal-directed, and time-limited intervention (the CoMMIT program) to clarify the importance of establishing reablement services as standard care. The CoMMIT program encourages participants to develop self-management skills to ensure adequate oral health, nutrition, physical activities, activities of daily living (ADL)/IADL, and social participation. A case study found that some older adults of the CoMMIT program could improve their self-management skills and then regain independence from long-term care services after attending the program [18]. This preliminary finding is promising, particularly since independence from long-term care services is rare among older adults with mild disability [18].

Therefore, we aimed to assess the efficacy of the CoMMIT program for older adults with mild disability. We hypothesized that the CoMMIT program plus standard care enhances these individuals’ independence from long-term care services over a three-month follow-up period compared with the provision of standard care alone.

## 2. Materials and Methods

### 2.1. Design

This trial is a parallel, 2-arm, randomized controlled, superiority trial for community-dwelling older adults with mild disability. The trial was conducted from 15 February to 30 November 2018 in Neyagawa, a local government area in Osaka, Japan. The study period was divided into a 1.5-month enrollment period, 5-month intervention period, and 3-month follow-up period (Appendix A). The study protocol was approved by the institutional review board of the Institute of Health and Economics (protocol number: H29-002) and Chiba University (protocol number: 2949), and the trial was prospectively registered on 15 February 2018 on UMIN000031329.

### 2.2. Setting

The community under study had approximately 67,000 inhabitants aged 65 years or older, who accounted for 28.1% of the community’s population. In 2017, due to their disability, 3565 of the inhabitants were certified as support-required level in long-term care insurance.

Since 2000, Japan has been implementing the public long-term care insurance system. The long-term care insurance is for people aged 45–64 years with disability arising from specific diseases (e.g., end-stage cancer) and for those aged 65 years or older with disability. Japan’s long-term care insurance is not only available to people with moderate to severe disability, but also to those with mild disability, which makes it a unique system. People with mild disability are allowed to choose their service providers on their own without the gatekeeping system. In addition, people with mild disability generally do not receive any reablement assessments to determine whether long-term care services would enhance their independence. Such a gatekeeping system and reablement assessment is common in England, Australia, and Denmark [19].

The certification process for disability assessment is based on a nationally standardized support needs assessment process to determine how many hours were required for caregiving [4]. Support-required level 1 is defined as a condition that requires from 25 to less than 32 min of long-term care. Support-required level 2 is defined as the need for 32 to less than 50 min of long-term care for individuals with normal cognitive functioning and whose disability may not progress within a short period. People with support-required-level certification are considered to have mild disability because most of them are independent in their ADL, but have partial difficulties in IADL [3]. In general, recipients should complete the recertification process 6 months after their initial certification and every 12 months after each subsequent certification.

### 2.3. Participants

From the enrollment list of long-term care insurance, we recruited participants through advertisements in the local newsletter and website as well as direct contact with service users. The enrollment period was from 15 February to 31 March 2018. Eligible participants were community-dwelling older adults aged 65 years or older who were certified as support-required level and reported current (i.e., prevalent or new) use of long-term care services. The criteria for exclusion were a physician’s diagnosis of dementia with a score of III or more on the Dementia Scale [20], physician’s diagnosis of end-stage cancer, and receipt of financial aid for treatment of an intractable disease [21]. The support-required level and dementia assessments were conducted as part of the certification process, and other assessments were conducted during enrollment. Further, the eligibility criteria were assessed by municipal officers. All participants provided written informed consent.

### 2.4. Randomization

Participants were assigned in a 1:1 ratio to receive either the CoMMIT program plus standard care or standard care alone. The Pocock–Simon randomization method was used to balance important covariates, including support-required level (level 1 vs. level 2), the current use of long-term care services (prevalent user vs. new user), and age group (65–74 years vs. 75–84 years vs. ≥85 years) using a randomization software [22,23]. Allocation concealment was achieved by onsite study coordinators, who assisted in the enrollment process, and an off-site coinvestigator (Y.O.), who ensured the random assignment of participants. Due to the nature of the intervention, participants and providers could not be blinded to the allocation.

### 2.5. Intervention

#### 2.5.1. Standard Care Group

Participants in the control group could use various long-term care services, including home-visit (e.g., home-help) services, commuting (e.g., day-care) services, short-stay services, facility services, at-home medical care management counseling, and rental of assistive equipment. In general, long-term care insurance pays for 90% of these service costs. Further, the participants of the control group had an opportunity to receive the CoMMIT program after the 3-month follow-up period.

#### 2.5.2. CoMMIT Program Plus Standard Care Group

Participants in the intervention group received the CoMMIT program plus standard care. The CoMMIT program is a community-based, multicomponent, multidisciplinary, individualized goal-directed, and time-limited (5-month) intervention program (Figure 1) [24]. The CoMMIT program focuses on enabling participants to return to a previous lifestyle. At the initial home-visit assessment, a care goal was formulated after discussion between participants and a rehabilitation specialist (i.e., a qualified physiotherapist or occupational therapist) with a care manager. The care goal was defined as a task that older adults were unable to perform due to their disability but wanted to be able to do in the near future (e.g., I want to join a chorus group). To determine whether the stated care goals matched the participants’ true desires, rehabilitation specialists conducted a comprehensive clinical assessment including ADL (e.g., bathing), IADL (e.g., preparing food), and social participation (e.g., talking with friends) using an original assessment sheet with items scored on a 3-point scale (“past independence”, “current independence”, and “desire to regain independence”).

The core component of the program comprises 12 commuting modules, which were delivered weekly in groups of a maximum of 11 participants. Each module lasted 2–3 h, including a 20-min motivational interview [25] to assess individualized goal attainment and encourage participants to regain self-management skills to maintain their oral health, nutrition, physical activities, ADL/IADL, and social participation based on the International Classification of Functioning, Disability and Health (ICF) [26]. In every module, rehabilitation specialists reviewed participants’ individualized goals to monitor their progress, assessed their daily physical activities and home-based training in the previous week, and encouraged their behavioral changes using an assessment sheet for self-management (see Appendix A). The initial goal setting could be modified according to participants’ desires. The care in each module was customized to achieve the individualized goal through improving body function, activities, and participation. For example, participants without oral health problems did not receive the oral health care in the module. In the module, participants received individualized training and/or supervision with homework to regain self-management skills (e.g., oral health care). In principle, only equipment that participants could use at home was used. Among the modules, the motivational interview is the most distinctive element that has not yet been accepted as standard care.

After 12 commuting modules, the care goal attainment was assessed by self-reported behavior (e.g., whether a participant joined a chorus group within a pre-defined limit or not) in a case conference by the care staff. The final commuting module involved the participants and care staff reviewing the previous modules, assessment of past and current difficulties, planning of future activities, and confirmation of situations in which participants should seek help as soon as possible, using a booklet for preventing long-term care needs (see Appendix A).

The CoMMIT program was delivered by four service providers. The program was primarily administered by 10 rehabilitation specialists. In addition, registered dietitians and/or dental hygienists administered the CoMMIT program to participants with dental and/or nutritional problems.

It is noted that rehabilitation specialists avoid physical contact during functional training, which allowed us to initiate the CoMMIT program without physicians’ directions. This is because rehabilitation specialists are required to obtain physicians’ directions to conduct physical and/or occupational therapy in the Japanese health and long-term care insurance system. Therefore, the CoMMIT program is legally different from physical and/or occupational therapy in the system. The long-term care insurance pays for 100% of these service costs as part of Commuting Service Type C in Community Support Projects.

### 2.6. Therapist Training and Quality Assurance

All the rehabilitation specialists in the intervention group received in-person training in a 4-day workshop (in total 10 hours) led by two supervisors who were occupational therapists with in-depth knowledge of the ICF [26]. Rehabilitation specialists learned about the CoMMIT program including the comprehensive assessments based on the ICF, motivational interviewing, and so on. Further, all cases in the initial home-visit assessment were monitored and supervised, and supervisors provided consultation to ensure that intervention fidelity was maintained during the intervention period.

### 2.7. Outcomes

The primary outcome of the study was independence from long-term care services. In this study, independence was defined as the nonuse of long-term care services during the 3-month follow-up period based on administrative claims data and interviews with care managers. Independence from long-term care services is a critical outcome for recipients, clinicians, and policymakers and is probably a proxy for disability improvement. The participants who did not use any long-term care services due to worsening health conditions (e.g., hospitalization or death) were considered to have no outcome events. Further, serious adverse events (i.e., hospitalization and death) were recorded during the 5-month intervention period plus the 3-month follow-up period. The primary outcome and serious adverse events were objective outcome measures. The complete list of secondary outcomes that were not used in this article is not shown because of space constraints; however, it included depression, quality-of-life, physical functioning, and physical activities, among others.

### 2.8. Sample Size Estimation

Our prespecified sample size was 600 participants (300 per group) with the following assumptions: (1) an incidence proportion of independence at 12.5% in the intervention group, (2) an incidence proportion of independence at 5% in the control group, (3) an expected dropout of 35% (for the per-protocol analysis), and (4) a 2-sided significance level of 5% and a power of 80%. The reasons for choosing the incidence proportions were based on previous records from another municipality and the clinical importance determined through personal communication with municipal officers. However, the study was terminated early due to slow enrollment and failed to achieve the planned statistical power.

### 2.9. Statistical Analyses

The primary analysis was performed on an intention-to-treat (ITT) principal. For this purpose, between-group comparisons of the primary outcome were performed using Chi-squared tests. The results of the primary outcome are presented as absolute differences with 95% confidence intervals (CIs) and additionally reported as incidence proportion ratios. Further, post-hoc subgroup analyses were performed on stratification variables (i.e., support-required level, use of long-term care insurance, and age group) and other clinically relevant variables (i.e., sex, dementia status, number of impaired ADL, and number of impaired IADL) with tests for interaction. The dementia status was assessed using a physician’s diagnosis of dementia having a score of I or II on the Dementia Scale [20]. The numbers of impaired ADL and IADL were assessed using the standardized assessment process (Appendix A) [4]. Further, the following variables were assessed at the time of the certification process: support-required level, dementia status, number of impaired ADL, and number of impaired IADL. Moreover, two sets of sensitivity analyses were performed by focusing on the participants who had received the allocated interventions at least once, that is, the full analysis set (FAS), and who had received more than half of the allocated interventions, that is, the per-protocol set (PPS). The maximum number of interventions was defined as 13 modules, including 12 commuting modules plus 1 review module in the intervention group and 5 months in the control group. All the analyses were conducted in R version 3.4.4 (R Foundation for Statistical Computing, Vienna, Austria), and a 2-sided *p* < 0.05 was considered statistically significant.

## 3. Results

### 3.1. Participants

A total of 375 participants were enrolled in the study and randomized to either the intervention group (*n* = 190) or control group (*n* = 185), as depicted in Figure 2. Among the participants randomized to the intervention group, 144 (75.8%) and 125 (65.8%) attended at least one module and seven modules, respectively. Thirty-two participants randomized to the intervention group dropped out before the start of the commuting modules because of worsening health conditions. Among those randomized to the control group, 173 (93.5%) and 168 (90.8%) received standard care for at least 1 month and 3 months, respectively. The baseline characteristics of the ITT, FAS, and PPS populations were similar across the assigned groups (Table 1). Further, the average numbers of visits for commuting services within the five-month intervention period were 26.1 and 19.9 visits in the intervention and control groups, respectively (Appendix A).

### 3.2. Outcomes

There were no missing data for the primary outcome and serious adverse events. Significant difference was observed in the incidence proportions of independence from long-term care services between participants who received the CoMMIT program or standard care alone in the ITT, FAS, and PPS populations. In the ITT population, the incidence proportions of independence were 11.1% in the intervention group and 3.8% in the control group (absolute difference: 7.3; 95% CI: 2.0–12.5; Figure 3). In the FAS and PPS populations, similar results were observed (Appendix A). Further, a much higher incidence proportion ratio was observed in the FAS population than in the ITT population (incidence proportion ratio: 19.7 for FAS vs. 2.5 for ITT), although the FAS and PPS populations showed similar ratios (incidence proportion ratio: 19.7 for FAS vs. 18.7 for PPS). The tests for interaction for exploratory subgroup analyses did not find any statistically significant difference in any of the subgroup categories. In addition, there was no difference in the risk of serious adverse events between the groups in the ITT, FAS, and PPS populations (Table 2).

## 4. Discussion

To our knowledge, this study is one of the largest trials to assess the efficacy of multicomponent interventions in enhancing the independence of community-dwelling older adults with mild disability. Our findings support the primary hypothesis that the CoMMIT program plus standard care is superior to the provision of standard care alone in enhancing older adults’ independence from long-term care services. This finding is consistent with the result of a previous study, according to which a 12-week-long home-based rehabilitation program reduced the demand for ongoing home-care services at the one-year follow-up [11]. These findings suggest that participants who received the CoMMIT program were more likely to regain self-management skills and thus were less likely to have needs for using long-term care services.

Our results showed that the effects of the CoMMIT program were not modified by dementia status, numbers of impaired ADL/IADL, and so on. Further, the risk of serious adverse events did not significantly differ between the participants who had received the CoMMIT program and those who had received standard care alone.

According to our study, the incidence proportion ratio was much higher in the FAS and PPS populations than in the ITT population. One reason for this difference is that the number of participants who could regain independence in the FAS and PPS populations was only one in the control group, whereas it was much higher than one in the intervention group. A high incidence proportion ratio might be misleading when an event is rare in one group but occurs more frequently in another group [27,28]. Therefore, readers should interpret our results of incidence proportion ratio with caution.

Further, the dropout rate immediately after enrollment was much higher in the intervention group (26%) than in the control group (6%). However, this finding may not reflect participants’ nonadherence to the CoMMIT program because the retention rate after the initiation of the intervention may be considered acceptable at 87%. Contrastingly, this result may reflect the coverage differences in the long-term care insurance system between recipients with mild disability and those with moderate-to-severe disability. Participants whose disability had progressed prior to the start of the commuting modules could not initiate the CoMMIT program. This is because the CoMMIT program only covers recipients with mild disability and not those with moderate-to-severe disability in the long-term care insurance system. By contrast, recipients can receive standard care regardless of their disability.

In the future, efforts should be expanded on integrating the CoMMIT program within standard care. Policymakers should ensure the quality of intervention delivery using several strategies, including the restriction of the number of service providers, specification of high-level certification criteria for service providers, and implementation of the pay-for-performance approach. Further, rehabilitation specialists should be aware of the importance of the motivational interview, which has not yet been accepted as standard care, in assessing individualized goal attainment and encouraging participants to regain their self-management skills.

This study has several strong points. First, the use of randomization software minimized selection bias due to inadequate sequence generation or inadequate allocation concealment [29]. Second, the use of objective outcome with the ITT principle minimized detection and attrition biases [29]. The use of objective outcome also minimized performance bias due to the knowledge of the assigned intervention [30]. Third, the large magnitude of effect with minimal heterogeneity (i.e., incidence proportion ratios of 2 or more in many subgroups) may allow us to grade the quality of our evidence as high [31].

Our study has several limitations. First, the participants were recruited from a single local government area; therefore, the generalizability of our findings remains unknown. Second, the early termination of the study considerably limited our study’s statistical power compared to the planned value, although our results were robust to various sensitivity and subgroup analyses. Third, the length of the follow-up period in our study was limited to 3 months, due to which long-term effects of the program could not be examined; we suggest that future studies should assess whether the CoMMIT program has long-term effects on older adults’ independence from long-term care services. Fourth, the definition of our primary outcome leads to limited comparability across previous studies; we suggest that future studies should use well-validated measures with extensive efforts to prevent missing outcome data. 

## 5. Conclusions

In conclusion, this study provides strong evidence that the CoMMIT program plus standard care is superior to the provision of standard care alone in enhancing the independence of older adults with mild disability from long-term care services. Our findings encourage future studies on the CoMMIT program to assess the durability and the efficacy of this program for other populations with mild disability.

## Figures and Tables

**Figure 1 ijerph-16-03954-f001:**
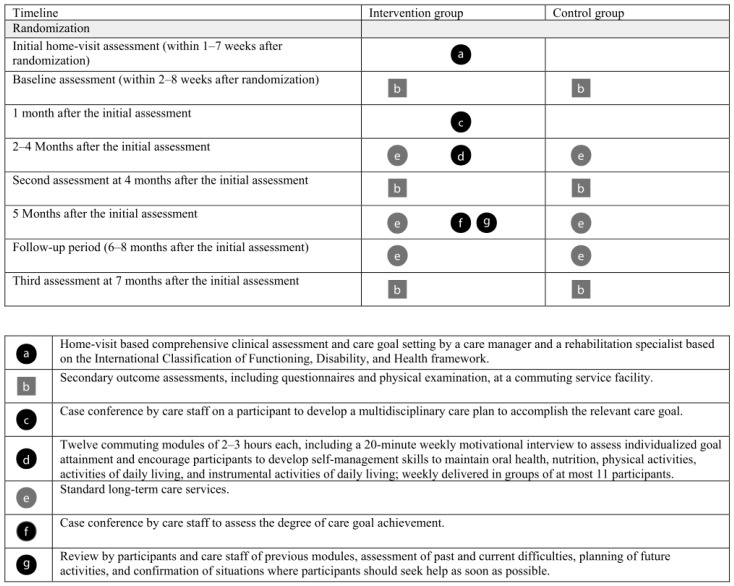
Timing and intervention elements. In the figure, squares reflect fixed components, circles reflect flexible components, objects highlighted in black represent the components included in the intervention group alone, and objects highlighted in gray represent the components included in both the intervention and control groups.

**Figure 2 ijerph-16-03954-f002:**
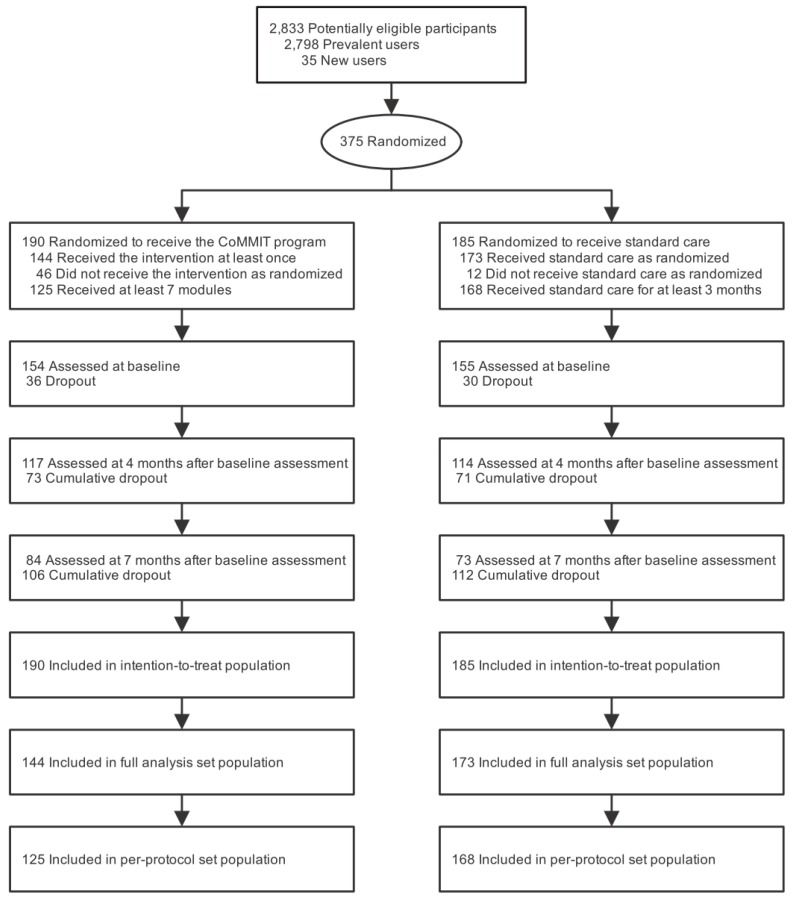
Flow diagram of the participants of the study.

**Figure 3 ijerph-16-03954-f003:**
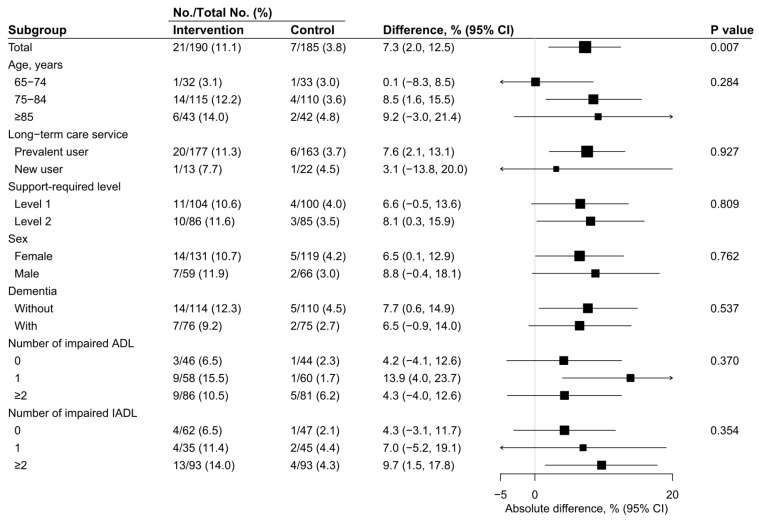
Effects of reablement on the independence from long-term care services of post-hoc subgroups in the intention-to-treat population. The *p*-values for subgroup comparisons correspond to the test for interaction. ADL, activities of daily living; CI, confidence interval; IADL, instrumental activities of daily living.

**Table 1 ijerph-16-03954-t001:** Baseline characteristics of participants receiving the long-term care insurance service randomized to intensive versus standard care.

Characteristics	ITT Population		FAS Population		PPS Population	
	Intervention Group (*n* = 190)	Control Group (*n* = 185)	Intervention Group (*n* = 144)	Control Group (*n* = 173)	Intervention group (*n* = 125)	Control Group (*n* = 168)
Age, median (IQR), years	80.0 (76.3–84.0)	80.0 (76.0–84.0)	80.0 (76.0–83.3)	80.0 (76.0–84.0)	80.0 (76.0–84.0)	80.0 (76.0–84.0)
Age group, y, No. (%)						
65–74	32 (16.8)	33 (17.8)	25 (17.4)	32 (18.5)	23 (18.4)	32 (19.0)
75–84	115 (60.5)	110 (59.5)	90 (62.5)	101 (58.4)	74 (59.2)	96 (57.1)
≥85	43 (22.6)	42 (22.7)	29 (20.1)	40 (23.1)	28 (22.4)	40 (23.8)
Sex, No. (%)						
Female	131 (68.9)	119 (64.3)	100 (69.4)	110 (63.6)	83 (66.4)	108 (64.3)
Male	59 (31.1)	66 (35.7)	44 (30.6)	63 (36.4)	42 (33.6)	60 (35.7)
Use of long-term care insurance, No. (%)						
Prevalent user	177 (93.2)	163 (88.1)	133 (92.4)	155 (89.6)	116 (92.8)	151 (89.9)
New user	13 (6.8)	22 (11.9)	11 (7.6)	18 (10.4)	9 (7.2)	17 (10.1)
Support-required level, No. (%)						
Level 1	104 (54.7)	100 (54.1)	80 (55.6)	94 (54.3)	66 (52.8)	93 (55.4)
Level 2	86 (45.3)	85 (45.9)	64 (44.4)	79 (45.7)	59 (47.2)	75 (44.6)
Dementia, No. (%)						
Without	114 (60.0)	110 (59.5)	88 (61.1)	102 (59.0)	78 (62.4)	99 (58.9)
I	51 (26.8)	49 (26.5)	36 (25.0)	48 (27.7)	31 (24.8)	46 (27.4)
II	25 (13.2)	26 (14.1)	20 (13.9)	23 (13.3)	16 (12.8)	23 (13.7)
Number of impaired ADL, No. (%)						
0	46 (24.2)	44 (23.8)	32 (22.2)	42 (24.3)	26 (20.8)	42 (25.0)
1	58 (30.5)	60 (32.4)	45 (31.2)	57 (32.9)	39 (31.2)	55 (32.7)
≥2	86 (45.3)	81 (43.8)	67 (46.5)	74 (42.8)	60 (48.0)	71 (42.3)
Number of impaired IADL, No. (%)						
0	62 (32.6)	47 (25.4)	48 (33.3)	44 (25.4)	42 (33.6)	42 (25.0)
1	35 (18.4)	45 (24.3)	31 (21.5)	41 (23.7)	24 (19.2)	41 (24.4)
≥2	93 (48.9)	93 (50.3)	65 (45.1)	88 (50.9)	59 (47.2)	85 (50.6)

ADL, activities of daily living; FAS, full analysis set; IADL, instrumental activities of daily living; ITT, intention-to-treat; IQR, interquartile range; No., numbers; PPS, per-protocol set. The FAS population refers to the participants who have received the allocated interventions at least once, and the PPS population refers to participants who have received more than half of the allocated interventions.

**Table 2 ijerph-16-03954-t002:** Risk of serious adverse events for different population sets.

Population	Number of Events/Total Number (%)		*p* Value
	Intervention Group	Control Group	
*ITT population*			
Any serious adverse event	20/190 (10.5)	16/185 (8.6)	0.659
Death	2/190 (1.1)	3/185 (1.6)	0.976
Hospitalization	20/190 (10.5)	15/185 (8.1)	0.530
*FAS population*			
Any serious adverse event	11/144 (7.6)	16/173 (9.2)	0.757
Death	1/144 (0.7)	3/173 (1.7)	0.749
Hospitalization	11/144 (7.6)	15/173 (8.7)	0.898
*PPS population*			
Any serious adverse event	10/125 (8.0)	13/168 (7.7)	1.000
Death	1/125 (0.8)	3/168 (1.8)	0.834
Hospitalization	10/125 (8.0)	12/168 (7.1)	0.959

FAS, full analysis set; ITT, intention-to-treat; PPS, per-protocol set. The FAS population refers to participants who have received the allocated interventions at least once, and the PPS population refers to those who have received more than half of the allocated interventions.

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
