# Peer review of "Effects of Reablement on the Independence of Community-Dwelling Older Adults with Mild Disability: A Randomized Controlled Trial"

_ijerph, 2019, doi:10.3390/ijerph16203954_

Round 1

Reviewer 1 Report

Thank you for got the possibility to review this manuscript and as followed there is some need of clarification to this manuscript.

Recommendations for Authors (will be shown to authors)

Yes

Can be improved

Must be improved

Not applicable

Does the introduction provide sufficient background and include all relevant references?

Definition;

1. mild disability

2. long-term care services (is it determinant that it will be long-term from day one?) and the level of these services.

Is the research design appropriate?

Yes

Are the methods adequately described?

Clearify the 12 community models – what do they contain and how are they provided?

What was done during the 2-3hours? Did the interview include goal assessments, encouragement of several activities.

What are the standardized assessment process?

How was the goal attainment conducted? Did you use an instrument?

Hasn´t any assessments/instruments been used? A lot is mentioned. E.g instrument for goal-setting, clinical assessments, questionnaire but no specific instruments are presented.

What assessments are included in the initial/second/follow-up sessions?

The components of the intervention elements needs to be defined.

The whole intervention with their different components needs to be clarified.

Who are the rehabilitation specialist? What profession do they have? And the supervisors who are experts? What is the difference between specialist and expert?

The workshops – where they 10h for 4 days, or 10h during 4 days? What did they include?

There are a lot of things that are mentioned but not explained.

Are the results clearly presented?

Was it enough to attend one module?

Are the conclusions supported by the results?

Can you please clarify the conclusion of the amount of participants who becomes independent, 3 vs 1 might not be considered to be “much higher” when a group consists of 190/185 participants.

The authors could to be more humble to their results.

How can you determine that motivational interviewing is importance and that it contributed to a positive outcome?

Can you please develop/elaborate how this study provides strong evidence that CoMMIT + standard is superior?

* English language and style

 Extensive editing of English language and style required

Moderate English changes required

 English language and style are fine/minor spell check required

* Some words and sentences needs to be reviewed

 I don't feel qualified to judge about the English language and style

Recommendations for Editors (will not be shown to authors)

If you answered yes to any of the following questions, please give details in the comments for editors box below.

Yes

No

Do you have any potential conflict of interest with regards to this paper?

 No

Did you detect plagiarism? There is no obvious signs of plagiarism.

No

Do you have any other ethical concerns about this study?

Ratings

High

Average

Low

No answer

* Originality / Novelty

X

* Significance of Content

X

* Quality of Presentation

X

* Scientific Soundness

X

* Interest to the readers

X

* Overall Merit

X

ADDITIONAL COMMENTS

Abstract

Lacking some intro to the topic

Page 2

Line 1 – clarify the sentence “the former’s efficacy…”

Line 12 – Its either results or not, a published study doesn´t have preliminary finding.

So the questions is: is these preliminary findings according to Ref 18 or is this another one?

Line 37+38 – what kind of advertisements were done?

Page 3

Line 9+10 – last sentence fits better in the method discussion and then you could also develop the reasoning why this is the case.

Page 4

2.7 Outcome:

Primary outcome: Is independence correlated to non-use of long-term care service? “independence from long-term care service”.

Where did you retrieved this information?

2.8 missing references for your choices.

Fig.1

What is a comprehensive clinical assessment and care goal setting? What does these include? What questions are included in the questionnaire and what type of exercise is included?

Care goal – who set these? The patient, the team + the patient, only the team? How did you set the degree of care goal? What was the aim with this? How was degree measured? How many goals was set?

How did you assess individual goals attainment?

Did you measure the “encouragement” – how do you know if you encouraged or not?

How was the assessment of past and current difficulties conducted? With which tools?

Table 2.

Differentiate the subheading “ITT population”/”FSA population”/”PPS Population”

Overall, it is difficult to judge the results and the discussion since the method section need some clarification.   

Author Response

Thank you for your constructive comments. Please find the attached file.

Reviewer 2 Report

Population aging has posed and will still pose challenges for countries,especially like Japan and China. This paper reports on analysis of data collected in Neyagawa, Osaka, Japan. The finding is helpful for the long-term care service program, the CoMMIT program and old adults with mild disability.

1.  I am asking for providing more detail about the CoMMIT program.

2.  An extensive body of literature about long-term care services and old adults for other countries is shortage, and it must be provided.

3.  The results section,on page 5, besides the basic statistical analysis, more detail about regression analysis should be expanded. What’s more important,the discussion needs to be expanded and tied back to the theoretical framework and hypotheses of the study.

Author Response

(The authors gave the same response as above.)
